# Critical appraisal of minimally invasive keyhole surgery for intracranial meningioma in a large case series

Jai Deep Thakur[1,2,3], Regin Jay Mallari[1], Alex Corlin[1], Samantha Yawitz[1], Amalia Eisenberg[1], John Rhee[1,2], Walavan Sivakumar[1,2], Howard Krauss[1,2], Neil Martin[1,2], Chester Griffiths[1,2], Garni Barkhoudarian[1,2], Daniel F. Kelly[1,2]*

**1** Pacific Neuroscience Institute, Providence Saint John's Health Center, Santa Monica, California, United States of America, **2** Saint John's Cancer Institute, Providence Saint John's Health Center, Santa Monica, California, United States of America, **3** University of South Alabama, Mobile, Alabama, United States of America

* dkelly@pacificneuro.org

**Data Availability Statement:** All relevant data are within the paper and its Supporting Information files.

## Abstract

### Background

Meningioma surgery has evolved over the last 20 years with increased use of minimally invasive approaches including the endoscopic endonasal route and endoscope-assisted and gravity-assisted transcranial approaches. As the "keyhole" concept remains controversial, we present detailed outcomes in a cohort series.

### Methods

Retrospective analysis was done for all patients undergoing meningioma removal at a tertiary brain tumor referral center from 2008–2021. Keyhole approaches were defined as: use of a minimally invasive "retractorless" approach for a given meningioma in which a traditional larger approach is often used instead. The surgical goal was maximal safe removal including conservative (subtotal) removal for some invasive locations. *Primary outcomes* were resection rates, complications, length of stay and Karnofsky Performance Scale (KPS). *Secondary outcomes* were endoscopy use, perioperative treatments, tumor control and acute MRI FLAIR/T2 changes to assess for brain manipulation and retraction injury.

### Results

Of 329 patients, keyhole approaches were utilized in 193(59%) patients (mean age 59±13; 30 (15.5%) had prior surgery) who underwent 213 operations; 205(96%) were skull base location. Approaches included: endoscopic endonasal (n = 74,35%), supraorbital (n = 73,34%), retromastoid (n = 38,18%), mini-pterional (n = 20,9%), suboccipital (n = 4,2%), and contralateral transfalcine (n = 4,2%). *Primary outcomes*: Gross total/near total (>90%) resection was achieved in 125(59%) (5% for petroclival, cavernous sinus/Meckel's cave, spheno-cavernous locations vs 77% for all other locations). Major complications included: permanent neurological worsening 12(6%), CSF leak 2(1%) meningitis 2(1%). There were

**Funding:** The author(s) received no specific funding for this work.

**Competing interests:** The authors have read the journal's policy and have the following competing interests: DFK receives royalties from Mizuho Inc. GB is a consultant for Vascular Technologies Inc. and Cerevasc Inc. outside of the current study. WS is a consultant for Stryker Corporation outside of the current study. This does not alter our adherence to PLOS ONE policies on sharing data and materials. There are no patents, products in development or marketed products associated with this research to declare.

no DVTs, PEs, MIs or 30-day mortality. Median LOS decreased from 3 to 2 days in the last 2 years; 94% were discharged to home with favorable 90-day KPS in 176(96%) patients. *Secondary outcomes*: Increased FLAIR/T2 changes were noted on POD#1/2 MRI in 36/213 (17%) cases, resolving in all but 11 (5.2%). Endoscopy was used in 87/139(63%) craniotomies, facilitating additional tumor removal in 55%. Tumor progression occurred in 26(13%) patients, mean follow-up 42±36 months.

## Conclusions & relevance

Our experience suggests minimally invasive keyhole transcranial and endoscopic endonasal meningioma removal is associated with comparable resection rates and low complication rates, short hospitalizations and high 90-day performance scores in comparison to prior reports using traditional skull base approaches. Subtotal removal may be appropriate for invasive/adherent meningiomas to avoid neurological deficits and other post-operative complications, although longer follow-up is needed. With careful patient selection and requisite experience, these approaches may be considered reasonable alternatives to traditional transcranial approaches.

## Introduction

Meningiomas are the most common primary brain tumor with almost 35,000 patients being diagnosed annually in the US [1], and although 85–90% are benign (WHO Grade 1), they frequently encase or become adherent to arteries, veins and cranial nerves, and have a propensity to invade multiple skull base compartments. While surgery remains first-line treatment for meningiomas, the approach used and aggressiveness of removal is highly location-dependent and influenced by other factors such as tumor invasiveness, tumor consistency, prior treatments and surgeons' philosophy. Innovative skull base surgery approaches developed in the early 1990s generally employed large incisions, extensive bone removal and promoted maximal tumor removal as the primary goal. However, results from multiple series using these approaches indicate that overly aggressive tumor removal can be associated with relatively high rates of permanent cranial nerve deficits and other neurological complications [2–5]. Furthermore, stereotactic radiosurgery and radiotherapy (SRS or SRT) have long-term control rates of 90% or higher for most recurrent or progressive WHO grade I meningiomas, and some primarily treated cavernous sinus (CS) meningiomas [6, 7]. Thus, over time, restoring or maintaining neurological function and quality of life have gained greater priority and the dictum of *maximal safe tumor removal* has gained wider acceptance [2, 8–11].

Minimally invasive surgical (MIS) techniques have been increasingly applied across multiple surgical subspecialties often with the aid of endoscopy [12–15]. Although the concept of minimally invasive brain tumor removal has been promoted for decades, it has remained controversial and not widely practiced. The term "keyhole surgery" was introduced 50 years ago by Donald Wilson in his 1971 technical note, "Limited exposure in cerebral surgery" [16]. Since the 1990s, the keyhole concept has been refined by application of modern micro-neurosurgical techniques and technology including low profile instrumentation, neuro-navigation and high-definition endoscopy. The keyhole concept emphasizes use of tailored and targeted approaches that limit brain exposure through strategically placed craniotomies, minimizing brain manipulation without static brain retractors, facilitated by gravity-assistance, and with

the ultimate goal of achieving maximal safe tumor removal [17–22]. Notably, in the last 20 years, the endoscopic endonasal approach has evolved into an accepted and commonly used minimally invasive route for many midline skull base pathologies including meningiomas, given its ability to facilitate tumor removal without brain retraction. As such, the endonasal craniotomy route to the ventral skull base should arguably be included in the keyhole definition and armamentarium, and by so doing provides a more comprehensive 360-degree keyhole paradigm for meningioma management [23–28].

For over a decade, we have used several minimally invasive approaches for brain tumors, particularly for skull base and parafalcine meningiomas that eliminate the need for fixed brain retraction by relying on gravity assistance and endoscopic visualization [27–35]. Except for our recent report on elderly meningioma patients and that by Burks et al, to our knowledge, there are no prior studies with over 40 patients treated with a minimally invasive keyhole paradigm for all intracranial meningiomas [26, 33]. Herein we report detailed outcomes of our experience with these approaches to intracranial meningiomas including extent of resection, complications, length of stay (LOS), performance status, readmissions, and resection and recurrence rates. We also quantify the degree of parenchymal brain injury from retraction and manipulation injury by measuring acute MRI changes, and to what extent was endoscopy helpful in transcranial approaches for achieving additional tumor removal. We then assess our outcomes in aggregate compared to those reported in prior publications in meningioma patients operated through traditional skull base approaches. With this comprehensive analysis we attempt to answer one overarching question: '*What is the benefit of keyhole surgery on neurological and overall surgical recovery and can such minimally invasive approaches be considered reasonable alternatives to traditional approaches*? Six illustrative case examples are provided in two videos (**S1 and S2 Videos**).

## Methods

### Patient population & data collection

After institutional review board approval (IRB# JWCI-19-1101), all patients at Saint John's Health Center, Santa Monica, CA, between January 2008 and January 2021 who underwent surgical removal of an intracranial meningioma were identified. Patient consent was not necessary as data was deidentified. All operations were performed by one of two neurosurgeons (DK, GB) and endonasal operations were performed with otolaryngology collaboration (CG). Data collection included prior treatments, tumor histopathology, size and location on MRI, endoscope usage, extent of resection, complications, length of stay, disposition, readmissions and long-term tumor control.

### MIS "keyhole" definition and approach selection

Keyhole approaches were defined as: *the use of a minimally invasive approach for a given tumor in which a traditional larger approach may be used instead*. These six approaches included the endoscopic endonasal, supraorbital, mini-pterional, retromastoid, suboccipital tentorial and transfalcine routes. This definition follows that detailed by Lan et al in their 2019 consensus paper on keyhole microneurosurgery and aligns with our definition from prior publications on keyhole surgery [21, 27, 28] (**Fig 1A & 1B, S1 Fig, S1 Table**). Non-keyhole approaches include convexity craniotomies, pterional, bifrontal and far-lateral approaches. **Table 1** describes the strategy for applying these 6 keyhole approaches and two **videos** provide 6 case examples. During the study period, traditional open skull base craniotomies including orbito-zygomatic, transpetrosal, translabyrinthine, transcondylar craniotomies were not used.

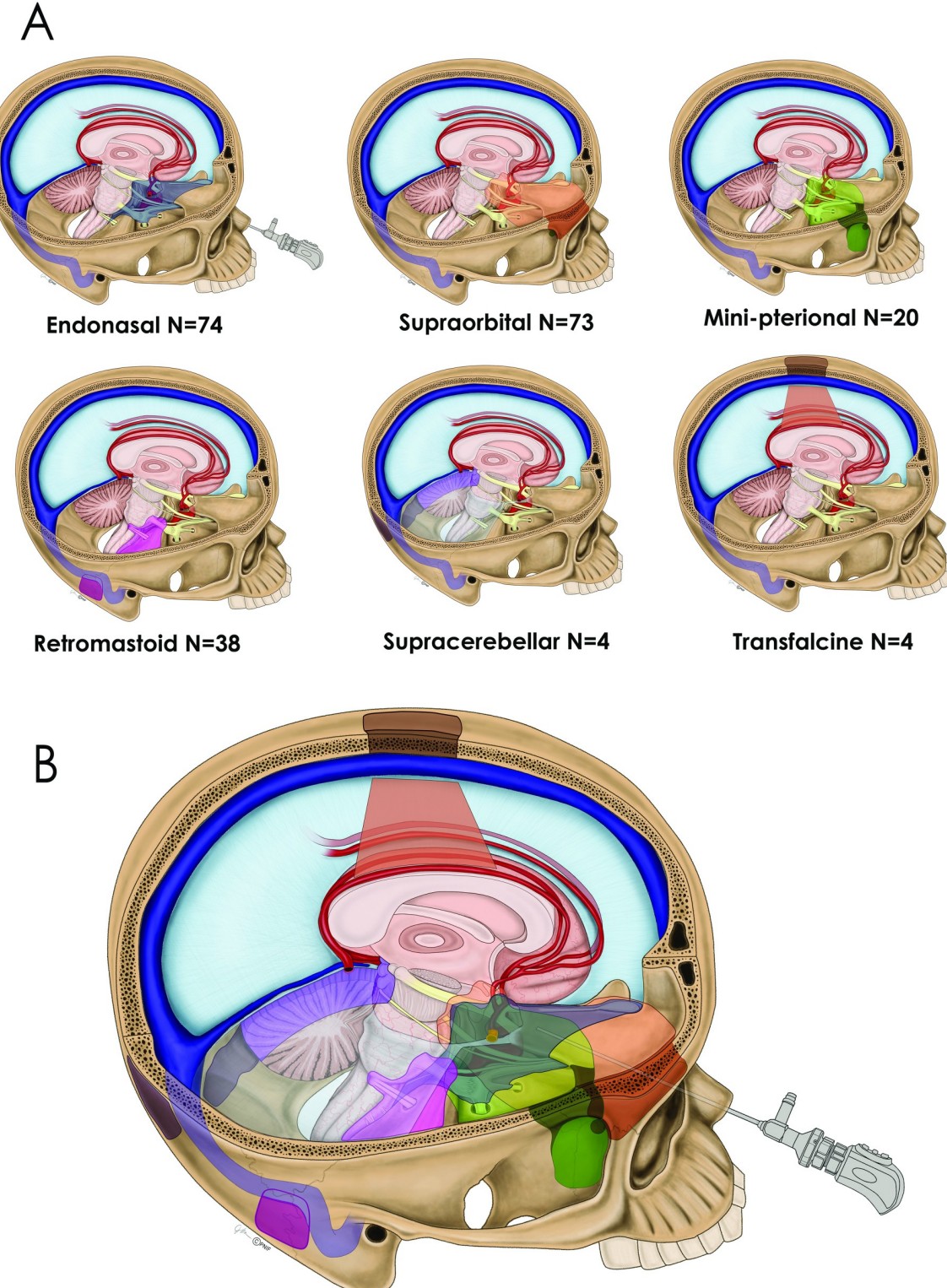

**Fig 1.** (1A) Drawing depicting 6 keyhole approaches for meningioma removal: endonasal, supraorbital, minipterional, retromastoid, suboccipital sitting gravity-assisted and transfalcine gravity-assisted, along with the total number of surgeries for each approach. (1B) A composite of the 6 keyhole approaches.

**Table 1. Surgical decision making for keyhole meningioma removal.**

| Meningioma Location | Factors for Surgical Decision Making | Approach Selection |
|---|---|---|
| **Olfactory Groove/Anterior Planum** | Olfaction preservation | **Supraorbital** |
| **Posterior Planum/ Tuberculum Sella** | 1. Proportion of tumor above the planum<br>2. Sellar depth<br>3. Tuberculum angle<br>3. Optic canal Invasion<br>4. Extent of tumor extension lateral to supraclinoid ICA<br>5. Maximal tumor diameter | Majority of tumor below planum, deep sella, steep (acute) tuberculum angle, minimal lateral extension, small size (under 3 cm): Favor **Endonasal**<br>Medial optic canal invasion: Favor **Endonasal**<br>Majority of tumor above planum, shallow sella, broad tuberculum angle, significant lateral extension, lack of medial optic canal invasion, larger size (over 3 cm): Favor **Supraorbital** |
| **Clinoidal** | Extension into middle fossa | Predominantly above the lesser wing: favor **Supraorbital** Predominantly within the middle fossa: favor **Mini-pterional** |
| **Sphenoid Wing** | Angle of Attack with respect to the Optic Chiasm and Supraclinoid Carotid | **Mini-pterional** |
| **Spheno-Orbital/Spheno Cavernous** | Angle of attack with respect to the optic chiasm and supraclinoid carotid artery | **Mini-pterional ± orbitotomy** |
| **Cavernous Sinus/ Meckel's Cave, Spheno-cavernous** | Surgical goal of decompression | **Endonasal** |
| **Petroclival/ CP Angle/ Foramen Magnum** | Clival and CP Angle component | If substantial petrous and posterior CP angle component posterolateral to CN VI: **Favor Retromastoid**<br>If substantial clival component more anterior: Favor **Endonasal ± Retromastoid** |
| **Tentorial** | Proximity to convexity | If away from convexity–**Suboccipital sitting gravity-assisted endoscopic-assisted or fully endoscopic** |
| **Falx** | Abutting primary motor or sensory cortex with overlying ipsilateral cortex | **Contralateral gravity-assisted trans- falcine endoscopic** |

Four patients who underwent a staged operation using one or more keyhole approaches are identified and each keyhole approach was considered as a separate outcome.

## Surgical goals, preparation and technique

A goal of maximal safe tumor removal was applied across all tumor locations. However, subtotal removal was typically the goal for three meningioma subtypes given their invasive nature and tendency to encase neurovascular structures: petroclival, CS/Meckel's cave, and sphenocavernous locations, as well as some meningiomas with prior surgery and/or radiotherapy [32]. Residual meningioma was deliberately left along neurovascular structures if too adherent or left in the skull base bone if deemed too infiltrative. Four patients had planned staged approaches for large meningiomas (maximal diameter >6 cm: endonasal debulking combined with a pterional, mini-pterional or retromastoid approach.

Total intravenous anesthesia (TIVA) was used to promote rapid emergence from anesthesia [34]. Preoperative tumor embolization was not used. Lumbar drains for CSF diversion were used infrequently and only early in the series. As recently described, all endonasal cases were performed fully endoscopically using a 0° 4-mm rigid endoscope initially; 30° and 45° endoscopes (Karl Storz-America, El Segundo, CA), are used at various stages of the procedure [27, 32]. A pedicled nasoseptal flap (or middle turbinate flap) is harvested in cases of tuberculum and planum meningiomas but not for all CS/Meckel cave meningiomas, depending upon CSF leak grade [27, 32, 33].

Except for the fully endoscopic sitting suboccipital approach, all transcranial keyhole craniotomies are performed initially with the microscope and then with endoscopy as needed for illuminating areas poorly seen with the microscope, and for endoscopic tumor resection. Endoscopy is typically performed with an assistant driving the endoscope. Patient positioning is critical to optimize gravity-assisted exposure and for endoscopy. Our use of the supraorbital

and mini-pterional approaches have been well-described [27, 34, 36]. The retromastoid approach is performed with the patient supine or lateral position, utilizing a short linear or curvilinear incision with approximately 2x3cm craniotomy [19]. The suboccipital approach is performed in the sitting or lateral position typically using an approximate 2x2cm craniotomy for tentorial meningiomas. The gravity-assisted transfalcine approach is performed specifically for parafalcine meningiomas that have a rind of edematous overlying eloquent cortex at risk from an ipsilateral approach. The patient is placed in lateral position with tumor side up and inclined in reverse Trendelenburg position with the vertex at 30–45 degrees [29], allowing for gravity-retraction and access from the unaffected contralateral side.

### Outcome and statistical analysis

The primary and secondary outcomes analyses detailed below attempt to answer the question of benefit of keyhole surgery approaches in terms of neurological and overall surgical recovery. Primary outcomes included extent of tumor resection, complications, hospital LOS, reoperations and readmissions within 90 days, 30-day mortality, KPS, and cranial (CN) outcomes. A favorable KPS at ≥90 days follow-up was defined as improved or unchanged from preoperative KPS. As previously published, resection rates were categorized as: gross total resection (GTR), near total resection (NTR ≥ 90% tumor resection), or subtotal resection (< 90%) [27, 28]. Simpson resection classification was not assessed given that most patients had invasive skull base meningiomas in which complete tumor resection and removal of involved bone and dura is typically not feasible [27, 28, 37].

Three secondary outcomes were included: 1) To quantify approach-related trauma of tumor removal, post-operative day #1 or 2 MRIs were independently assessed by a neuroradiologist (JR) for new FLAIR/T2 signal changes in the peritumoral and surgical approach areas [27]. New FLAIR/T2 change in the axial plane around the resection area or approach trajectory was quantified by maximal diameters. Persistence or resolution was documented on 3-month postoperative MRI. 2) To assess the utility of endoscopic visualization for transcranial operations, as previously published, it was determined if endoscopy facilitated additional tumor removal [27, 34] 3) To assess tumor progression or recurrence, sequential postoperative MRIs and need for additional treatment (repeat surgery and/or radiotherapy) were quantified for each patient.

The statistical comparison of the mean in the data amongst different groups was performed using ANOVA and independent Student's t-test. Univariate analysis was done using Chi-square or Fisher's exact test when applicable. Binomial multivariate analysis was done using logistic regression analysis and p-values less than 0.05 were considered statistically significant. Tumor control data for testing binomial variables were plotted using Cox Regression and Kaplan-Meier analysis.

## Results

### Demographics

Of 329 patients, keyhole approaches were used in 193 (59%) (74% female, mean age 59±13, mean follow up 42±36 months; WHO Grade I: 173/193(89.6%), Grade II: 19/193(9.8%), Grade III: 1/193(0.6%)); 32(9.7%) had prior radiation. The 213 keyhole operations included 205 (96%) for skull base location and 47(22%) were in patients with prior surgery. Operative approaches used included: endonasal (n = 74), supraorbital (n = 73), retromastoid (n = 38), mini-pterional (n = 20), sitting suboccipital (n = 4), and contralateral gravity-assisted transfalcine (n = 4) (**Table 2**). Four patients had planned staged operations using two approaches for large meningiomas (endonasal combined with a pterional n = 1, supraorbital n = 1, mini-

**Table 2. Meningioma characteristics, resection rates & surgical parameters for 213 keyhole operations.**

| | Endonasal (n = 74) | Supraorbital (n = 73) | Mini-pterional (n = 20) | Retromastoid (n = 38) | Suboccipital (n = 4) | Transfalcine (n = 4) | Total (n = 213) |
|---|---|---|---|---|---|---|---|
| **GTR** | 17 (23%) | 34 (46.6%) | 10 (50%) | 19 (50%) | 2 (50%) | 3 (75%) | 85 (40%) |
| **GTR/NTR** | 22 (29.7%) | 54 (74%) | 14 (70%) | 28 (73.7%) | 3 (75%) | 4 (100%) | 125 (58.7%) |
| **Redo Surgery (had prior surgery)** | 24 (32.4%) | 16 (21.9%) | 3 (15%) | 3 (7.9%) | 1 (25%) | 0 | 47 (22.1%) |
| **Invasion to CS/MC/Orbit /ITF** | 40 (54%) | 15 (21%) | 10 (50%) | 9 (24%) | 0 | 0 | 74 (35%) |
| **Median Skull Base Compartments Occupied** | 2 | 2 | 2 | 2 | 1 | 1 | 2 |
| **Mean Max Tumor Diameter (mm)** | 29.1 ± 13.2 | 28.5 ± 13.0 | 29.7 ± 14.1 | 32.1 ± 13.5 | 26.5 ± 12.5 | 40.0 ± 21.2 | 29.7 ± 13.4 |
| **Use of Endoscopy** | 74 (100%) | 54/73 (74%) | 5/20 (25%) | 21/38 (55%) | 3/4 (75%) | 4/4 (100%) | 161/213 (76%) |
| **New/Worsened FLAIR/T2 Changes (mean diam, mm)** | 2/74 | 11/73 | 4/20 | 18/48 | 1/4 | 0/4 | 36/213 |
| | 2.7% | 15.1% | 20.0% | 37.5% | 25% | 0 | (16.9%) |
| | (2.5mm) | (6.9 mm) | (9.5 mm) | (8.1 mm) | (11 mm) | (NA) | (7.67 mm) |
| **Persistent FLAIR Changes at 3 months or more postop** | 1/74 (1.4%) | 2/73 (2.7%) | 2/20 (10%) | 6/48 (12.5%) | 0 | 0 | 11/213 (5.2%) |
| **Median LOS** | 3 | 3 | 2 | 2 | 3 | 2 | 3 |

Abbreviations: GTR: gross total resection, NTR: near total resection, CS: cavernous Sinus, MC: Meckel's cave. LOS: length of stay, Max: maximum

pterional n = 1 or retromastoid approach n = 1). Lumbar drains were used in 5/74(7%) endonasal operations all prior to 2013, and 1/139 (0.7%) transcranial operation in 2017.

## Primary outcomes

**Resection rates by approach and tumor location.** As shown in **Tables 2 and 3**, GTR was achieved in 85(40%) operations while GTR/NTR (>90% resection) was achieved in 125(59%) operations. By meningioma location, GTR/NTR rates were highest for frontal fossa, parafalcine tentorial, olfactory groove and planum meningiomas (N = 158, range 84–100%), and lowest for invasive spheno-cavernous, CS/Meckel's cave and petroclival locations (N = 55, range 5–6%) (**Table 3**). Excluding these invasive subgroups (petroclival, CS/MC and spheno-cavernous), GTR/NTR was achieved in 114/136 (84%) first-time operations. Dense adherence to neurovascular structures was associated with 36% GTR/NTR versus 100%, without dense adhesions (p<0.001) (**S2 Table**). All patients with STR/NTR had at least one risk factor (prior surgery, radiotherapy, invasion cavernous sinus/Meckel's cave/infratemporal fossa/orbit).

**Clinical outcomes and complications.** Major complications occurred in 23/193 (11.9%) patients (**Table 4**). Permanent neurological worsening occurred in 12(6%) patients including 4 with strokes and 8 with cranial nerve injuries (**S3 Table**). Reoperations were needed in 10(5%) patients, (two for CSF leaks); 2(1%) patients had meningitis No patients developed perioperative DVT, PE or MI. Three patients all with multiple prior surgeries, died of disease progression at 80, 229, and 3508 days after their last operation.

**Discharge to home, functional outcomes and readmissions.** Of 213 operations, 201 (94%) resulted in patients being discharged to home and 12(6%) to skilled nursing facility or rehabilitation (**Table 4**). Median LOS was higher for patients discharged to rehabilitation or skilled nursing facility versus home (5 vs 2, p = 0.012). Over 11 years, median LOS decreased: 3 days (first 71 cases), 3 days (middle 71 cases) and 2 days (last 71 cases), p = 0.013. Longer LOS (LOS≥ 4) was associated with major complications (22% vs 7%, p = 0.005), discharge to rehabilitation or nursing facility (67% vs. 28%, p = 0.008). Ninety-day readmissions occurred in 6 (2.8%) cases, including 4 who required surgery.

**Table 3. Resection rates in 213 keyhole approaches by meningioma location, invasiveness & use of endoscopy.**

| Tumor Location (N) | Overall GTR/NTR (n = 213) | First-time Surgery GTR/NTR (n = 166) | Overall GTR (n = 213) | First-time Surgery GTR (n = 166) | Use of Endoscopy (n = 213) |
|---|---|---|---|---|---|
| Frontal Fossa (8) | 8 (100%) | 8/8 (100%) | 8 (100%) | 8/8 (100%) | 7 (88%) |
| Parafalcine (4) | 4 (100%) | 4/4 (100%) | 3 (75%) | 3/4 (75%) | 4 (100%) |
| Tentorial (18) | 17 (94%) | 17/17 (100%) | 13 (72%) | 13/17 (77%) | 12 (67%) |
| Olfactory Groove (14) | 12 (86%) | 11/12 (92%) | 9 (64%) | 8/12 (67%) | 14 (100%) |
| Planum Sphenoidale (12) | 10 (84%) | 8/10 (80%) | 4 (33%) | 3/10 (30%) | 9 (75%) |
| Clinoidal (19) | 14 (74%) | 13/15 (87%) | 8 (42%) | 8/15 (53%) | 12 (63%) |
| Tuberculum Sellae (39) | 28 (72%) | 26/31 (84%) | 22 (56%) | 22/31 (71%) | 33 (85%) |
| Cerebellopontine Angle (23) | 16 (70%) | 14/21 (67%) | 9 (39%) | 8/21 (38%) | 14 (61%) |
| Sphenoid Wing (16) | 10 (62.5%) | 10/15 (67%) | 7 (44%) | 7/15 (47%) | 7 (44%) |
| Spheno-orbital (5) | 3 (60%) | 3/3 (100%) | 1 (20%) | 1/3 (33%) | 1 (20%) |
| **Total** | **122/158 = 77%** | **114/136 = 84%** | **84/158 = 53%** | **81/136 = 60%** | |
| Petroclival (17)* | 1 (6%) | 1/9 (11%) | 0 (0%) | 0/9 (0%) | 14 (82%) |
| CS/MC (19)* | 1 (5%) | 0/10 (0%) | 0 (0%) | 0/10 (0%) | 18 (95%) |
| Spheno-cavernous (19)* | 1 (5%) | 1/11 (10%) | 1 (5%) | 1/11 (10%) | 16 (84%) |
| **Total** | **3/55 = 5%** | **2/30 = 7%** | **1/55 = 2%** | **1/30 = 3%** | |
| **TOTAL** | **125 (59%)** | **116 (70%)** | **85 (40%)** | **82 (49%)** | **161 (76%)** |
| **Dense adherence to critical neurovascular structures** | | | | | |
| Yes (N = 138) | 50 (36%) | | 20 (15%) | | |
| No (N = 75) | 75 (100%) | | 65 (87%) | | |
| p-value | **p<0.001** | | **p<0.001** | | |

Abbreviations: GTR: gross total resection, NTR: near total resection, CS: cavernous Sinus, MC: Meckel's cave.

*These 3 locations represent a subset of 55 invasive skull base meningiomas in which conservative (subtotal) removal was the surgical goal. Of these 55 cases; 7 patients had tumor progression treated with repeat surgery and/or SRS/SRT.

Of 17 operations for petroclival meningioma (in 13 patients), 14 were approached via the endonasal route for conservative debulking and 3 via the retromastoid route; 13/17 (76%) operations and 9/13 (69%) patients had meningiomas that extended into multiple compartments including CS, Meckel's cave, and/or sellar/suprasellar areas, 8/13 (61%) had prior surgery and 7 (54%) had prior radiation.

Of 184/193 (95%) patients with available follow-up, favorable 90-day KPS was noted in 176 (96%): improved in 126/176(72%) and stable in 50/176 (28%). Mean KPS for the entire cohort improved postoperatively from 72.3 to 82.4 (p<0.001).

## Secondary outcomes

**Acute MRI changes in region of approach.** Of 213 operations, POD 1 or 2 MRIs showed regional increase in FLAIR/T2 signal in 36(16.9%), highest in the retromastoid cohort (37.5%) (Table 2). FLAIR/T2 averaged 8±5 mm in maximal diameter and completely resolved in all but 11(5.2%) cases on follow-up MRI. No patients with increased FLAIR had attributable neurological deficits to their FLAIR changes.

**Utility of endoscopy.** Endoscopy was used in 161(76%) keyhole operations, including all 74 endonasal cases and 87/139(63%) transcranial cases (Table 2). Of 87 endoscope-assisted transcranial cases, in 48(55%) it facilitated additional tumor removal.

**Long-term tumor progression or recurrence.** Mean follow-up for the 193 patients was 41±36 months. Recurrence after GTR was seen in 1(1.2%) patient, while progression of residual tumor occurred in 26(24.1%) patients, for an overall recurrence/progression rate of 13.9% (27/193). As per Kaplan-Meier analysis, mean time to recurrence or progression

**Table 4. Major and minor complications, readmissions, reoperations and discharge status in 193 patients undergoing 213 keyhole operations for meningioma.**

| | |
|---|---|
| *Major Surgical Complications (n = 23 patients)* | **25** |
| **Permanent Neurological Worsening** | **12(6%)** |
| • Stroke | 4(2%) |
| • New or Worsening Cranial Nerve Dysfunction | 8 (4%) |
| **Transient Neurological Worsening** | **1(0.5%)** |
| • Persistent seizures with transient hemiparesis | 1 |
| **Reoperations** | **10(5%)** |
| • Delayed hematoma evacuation | 2 |
| • Reoperation for residual tumor (same admission) | 2 |
| • Reoperation for residual tumor (readmission) | 1 |
| • CSF leak repair | 2 |
| • Revision of sellar reconstruction (no CSF leak) | 2 |
| • Epistaxis needing surgical intervention | 1 |
| **Meningitis** | **2 (1%)** |
| **Total Major Complications by Operation** (p = 0.45) | |
| • Redo-operation | 7/47 (15%) |
| • First-time operation | 18/166 (11%) |
| *Minor Complications* | |
| • Sinusitis | 3 |
| • Mucocele | 1 |
| • Forehead numbness | 11 |
| • Frontalis paresis | 7 |
| • Frontalis palsy | 2 |
| • Delayed wound dehiscence | 1 |
| • Hardware malposition | 1 |
| *Systemic Complications* | 2 (1%) |
| • Aspiration Pneumonia | 1 |
| • UTI | 1 |
| • DVT/PE/ MI | 0 |
| *Delayed Radiation Induced Optic Neuropathy* | 1 |
| **Discharge to Home** | 201/213 (94%) |
| **Readmissions Requiring Surgical Intervention (n = 4/213)** | 2% |
| • Residual tumor needing more surgery | 1 |
| • Delayed hematoma needing surgery | 1 |
| • CSF leak repair | |
| • Epistaxis | 1 |
| **Readmissions Managed Medically (n = 2/213)** | 1% |
| • UTI, Atrial fibrillation | |
| • Hyponatremia | 1 |

\* One patient had both CSF leak and meningitis; one patient who had stroke had a multiply recurrent meningioma with prior surgery and RT and was the only mortality in the series

was 23.7 ± 27.3 months (range 3–94 months). Of these 27 patients, 4 had repeat surgery only, 11 SRT only, 2 SRS followed by another surgery and 10 were observed or medically managed.

## Discussion

### MIS philosophy applied to meningiomas

MIS techniques aim to minimize collateral damage to normal tissues while achieving the surgical goal, be it tumor resection, valve replacement or disc replacement [12, 13, 15, 38, 39]. Striking that balance while being cost-effective and maximizing patients' quality of life is the ultimate "sweet spot" of any surgical procedure [38, 40]. We propose that the present series of MIS approaches and prior series from our group and others for intracranial meningioma are a step forward for neurosurgery and endoscopic skull base surgery, demonstrating the potential effectiveness of this keyhole paradigm [19–21, 28, 34]. In attempting to answer our central question, we suggest these results show a potential benefit of keyhole meningioma surgery on neurological and overall patient recovery, and thus offer a reasonable and in some instances, a preferred alternative to traditional approaches.

This evolution of keyhole surgery is in part a result of cross-specialty collaboration. The adoption of the endoscope into transsphenoidal surgery that began in the 1990s with our colleagues in otorhinolaryngology, has transformed not only pituitary surgery but also the entire field of skull base surgery. Most pituitary adenomas are now removed through an endoscopic endonasal approach, and as we show here, many midline skull base meningiomas can be removed via the endonasal route [27, 41, 42]. Furthermore, the endoscope was used in almost two thirds of transcranial keyhole approaches and helped facilitate additional tumor removal in 55% of these operations.

Keyhole meningioma surgery aims to limit brain exposure and manipulation, accessing tumors through smaller strategically placed craniotomies, including ventral skull base craniotomies via the endoscopic endonasal approach, without static brain retractors, facilitated by gravity-assistance, low profile instrumentation and endoscopy, with the goal of achieving maximal safe tumor removal [17–21]. Despite the benefits of visualization, there is ample evidence that fixed brain retractors can cause acute and lasting brain injury [17, 43]. Recent reports highlight the potential for retraction injury and associated complications with traditional craniotomies for tuberculum and planum meningiomas [44–46]. A major benefit of keyhole retractorless and gravity-assisted endoscopic approaches and the endoscopic endonasal approach may be in less brain exposure and parenchymal manipulation [17, 18, 20, 21, 27, 46, 47]. This advantage was evidenced by the absence of early postoperative FLAIR/T2 changes along the surgical corridor in 82% of patients in this series including 100% of endonasal approaches; for 36 patients with FLAIR/T2 increases, these were small and resolved in all but 5% of patients.

For a neoplasm as diverse in location and invasiveness as meningioma, the surgical team should be facile working through multiple surgical corridors with both microscope and endoscope (see Table 1). Using TIVA anesthesia, smaller scalp and muscle incisions, minimal monopolar cautery, and focused craniotomies to minimize brain exposure, appears to promote rapid healing, less post-operative pain and a greater willingness and ability for patients to mobilize and leave the hospital [34, 35]. Adoption of MIS techniques have been a key component of success in enhanced recovery protocols in other surgical subspecialties which we are adopting as well in all of our brain tumor patients [34, 48]. Having reliable skull base closure techniques especially for endonasal, supraorbital and retromastoid routes where bony sinus and mastoid air cell entry frequently occurs is also essential to avoid CSF leaks and meningitis. This preventative strategy includes liberal use of abdominal fat grafts to obliterate sinus or air cell entry after craniotomy and a graded repair approach for skull base reconstruction including fat grafts and nasoseptal flaps in endonasal surgery without lumbar CSF drainage [33, 36, 49].

## Complication rates, functional recovery and LOS compared to prior reports

To be valid and to show benefit in terms of neurological and overall surgical recovery, the MIS keyhole concept should yield low rates of new neurological deficits, CSF leaks and high rates of functional outcomes. Our results in terms of infrequent complications, short LOS and improved post-operative KPS scores support this approach. New cranial neuropathy was observed in only 4% of patients. Lumbar drains were used in only 3% of operations, while post-operative CSF leak rate and meningitis rates were only 1%. There were no cases of perioperative DVT, PE, MI or 30-day mortality. The absence of thromboembolic events is likely due in part to our high patient functionality with few neurovascular complication, early ambulation and limited perioperative narcotic use, and compares favorably to the 2.7–4.1% incidence of PE/DVT recently published [50, 51].

Prior studies and our experience demonstrate that complications impact quality of life, lengthen hospital stay, increase costs and often require reoperations and readmissions [29, 36, 38–40]. Compared to our outcomes of 3-day median LOS, 94% discharged to home, and 7% 90-day readmission rate, and no 30-day mortality, recent reports encompassing all intracranial meningiomas have documented LOS for skull base and other meningioma patients ranging from 4–11 days, discharge to home ranging from 70–83.4%, 90-day readmission rates of 9.2–17.9%, and 30-day mortality ranging from 0–5.4% [52–56]. This comparison is notable, given our series is comprised predominantly (96%) of skull base meningiomas (which are generally considered to be of higher complexity and risk profile than convexity meningiomas), while these 5 series include all intracranial meningiomas. A rapid complication-free recovery also benefits those patients who may have more aggressive (WHO Graded 2 or 3) or previously-treated meningiomas who may need to begin adjuvant therapy shortly after surgery [37].

In our recent publication on streamlining brain tumor care during the COVID-19 pandemic, we highlighted the importance and utility of an all-encompassing minimally invasive 360-degree approach to brain tumor surgery (for all tumor types), as a foundation for being able to not monitor many patients in the ICU and allowing for early hospital discharge, typically on POD#1 or #2 [35]. In this case control series of 293 patients, from pre-pandemic to pandemic, ICU utilization decreased from 54% to 29% of operations (p<0.001) and hospital LOS≤1 day increased from 12.2% to 41.4%, p<0.001, respectively. We believe that more routine and rigorous patient education on early discharge, recovery room assessment for non-ICU admission and earlier mobilization, layered upon a foundation of MIS, TIVA anesthesia and early post-operative imaging contributed to these significant and favorable trends (including our median LOS of 2 days in the last tertile of patients in this meningioma cohort).

## Balancing goals of maximal tumor removal and complication avoidance

Perhaps the most serious critique of keyhole meningioma surgery is that ultimately the patient is not well-served because overly conservative tumor removal leads to the eventual need for repeat surgery, radiosurgery, or possibly both. This issue is the essence of the second part of our central question: *'can keyhole approaches be considered reasonable alternatives to traditional approaches*?*'* Our overall meningioma resection rates for all locations except petroclival meningiomas are comparable to prior reports [8, 9, 44–46, 56], and our progression/recurrence rate of 14% is similar to prior reports (almost all in patients who underwent NTR or STR), although this rate will undoubtedly increase with longer follow-up [8, 57–62]. First-time resection rates for anterior cranial fossa and suprasellar meningiomas in the literature vary widely, ranging from 63% to 100%, with recurrence/progression rates ranging from 2% to 14% [19, 63, 64]. Considering petroclival meningiomas, we did not achieve GTR in any of 17

operations and had one new CN deficit (6%). In multiple series of petroclival meningiomas approached through traditional skull base approaches including the retromastoid approach, the GTR rate ranged from 21%-76% but permanent CN and other neurovascular complications ranged from 22%-54% [2–5, 65–71]. Collectively, these reports indicate that overly aggressive attempts at GTR, will likely be associated with a relatively high rate of permanent neurovascular morbidity and lower quality of life for many patients; thus we prefer a more conservative surgical approach for such invasive skull base meningiomas as other groups have also recommended [8, 10, 57–61, 72, 73].

A growing collective experience places functional preservation as a higher priority than GTR resection, as highlighted by recent reports [2, 8, 9, 37]. The beneficial impact of this approach for a given patient is a greater likelihood of no new postoperative neurological deficits and preserved or improved QOL while acknowledging that for many patients their meningioma becomes a chronic illness that warrants long-term monitoring with a higher likelihood of tumor progression and need for additional surgery, SRS/SRT or possible medical therapies in the years after non-GTR resection [8, 10, 57–61, 72, 73].

We highlight in Table 3 and S2 Table the factors most commonly responsible for not achieving a GTR which include invasiveness into the skull base and dense adherence to critical neurovascular structures. Heeding such anatomical realities accounts in large part for our low neurovascular complication rate but relatively high residual tumor rate for some locations. Overall, our results and those of other centers, seem to indicate a keyhole paradigm applied to meningiomas is a reasonable alternative to traditional skull base approaches, given our low neurovascular and systemic complication rates and short LOS, although further validation with longer follow-up at multiple centers is needed [19–22, 27, 28, 42].

## Study limitations and bias

The major limitation of this study is its retrospective nature, heterogenous meningioma patient population, and our selection bias for using these 6 keyhole approaches without a comparison cohort of patients treated with traditional skull base approaches or other relatively new minimally invasive approaches such as the endoscopic transorbital route [74, 75]. We specifically did not compare clinical outcomes in the keyhole cohort to the 136 (41%) non-keyhole patient cohort. Also, the follow-up in our patients averaged 44 months which is relatively short. Longer follow-up is necessary to assess the efficacy of this approach more fully and determine how many patients who underwent NTR or STR ultimately need radiotherapy or additional surgery.

## Conclusions

A minimally invasive keyhole paradigm applied to skull base and other select meningiomas can yield reasonable tumor resection rates with low rates of neurovascular and systemic complications, short LOS and high rates of functional outcomes. While traditional skull base approaches remain useful and effective, we suggest that with increasing experience, transcranial and endonasal keyhole approaches can be considered part of the surgical armamentarium for many intracranial meningiomas.

## Supporting information

**S1 Fig. Six keyhole approaches animated GIF.** The corresponding GIF animation of the 6 approaches illustrated in Fig 1.
(GIF)

**S1 Table. Keyhole and traditional approaches used for skull base and non-skull base meningiomas.** A breakdown of number of cases in which keyhole and traditional approaches were used for skull base and non-skull base meningiomas.
(DOCX)

**S2 Table. Multivariate regression analysis for tumor resection and progression.** Preoperative factors limiting GTR and factors predicting progression are analyzed using multivariate binomial and cox regression analysis.
(DOCX)

**S3 Table. Cranial nerve outcomes in 193 patients undergoing Keyhole meningioma removal.** A breakdown of the cranial nerve outcomes and recovery for patients undergoing keyhole meningioma surgery.
(DOCX)

**S1 Video. Anterior cranial fossa case examples.** Illustrative case examples of 3 anterior cranial fossa meningiomas: 1) tuberculum sella meningioma approached via endoscopic endonasal route, 2) olfactory groove meningioma approached via supraorbital route; 3) clinoidal meningioma approached via supraorbital route.
(DOCX)

**S2 Video. Three meningioma illustrative case examples.** Illustrative case examples of 3 meningiomas: 1) petroclival meningioma approached via retromastoid route, 2) tentorial meningioma approached via suboccipital sitting position route; 3) falx meningioma approached via transfalcine gravity-assisted route.
(DOCX)

**S1 Dataset.**
(XLSX)

## Acknowledgments

The authors would like to acknowledge the support of Pacific Neuroscience Institute Foundation and Saint John's Health Center Foundation for their support. We would also like to acknowledge Josh Emerson for providing the artwork for Fig 1.

## Author Contributions

**Conceptualization:** Jai Deep Thakur, Regin Jay Mallari, Amalia Eisenberg, Walavan Sivakumar, Howard Krauss, Neil Martin, Chester Griffiths, Garni Barkhoudarian, Daniel F. Kelly.

**Data curation:** Regin Jay Mallari, Alex Corlin, Samantha Yawitz, Amalia Eisenberg, John Rhee.

**Formal analysis:** Jai Deep Thakur, Regin Jay Mallari, John Rhee, Garni Barkhoudarian, Daniel F. Kelly.

**Investigation:** Jai Deep Thakur, Regin Jay Mallari, Walavan Sivakumar, Howard Krauss, Neil Martin, Chester Griffiths, Garni Barkhoudarian, Daniel F. Kelly.

**Methodology:** Jai Deep Thakur, Regin Jay Mallari, Walavan Sivakumar, Howard Krauss, Neil Martin, Chester Griffiths, Garni Barkhoudarian, Daniel F. Kelly.

**Project administration:** Garni Barkhoudarian, Daniel F. Kelly.

**Resources:** Garni Barkhoudarian, Daniel F. Kelly.

**Software:** Regin Jay Mallari.

**Supervision:** Garni Barkhoudarian, Daniel F. Kelly.

**Validation:** Jai Deep Thakur, Regin Jay Mallari.

**Visualization:** Jai Deep Thakur, Regin Jay Mallari.

**Writing – original draft:** Jai Deep Thakur, Regin Jay Mallari, Walavan Sivakumar, Daniel F. Kelly.

**Writing – review & editing:** Jai Deep Thakur, Regin Jay Mallari, Amalia Eisenberg, John Rhee, Walavan Sivakumar, Howard Krauss, Neil Martin, Chester Griffiths, Garni Barkhoudarian, Daniel F. Kelly.

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
