## [Decision Letter · Decision Letter 0]

16 Mar 2022

PONE-D-22-03249Critical appraisal of minimally invasive keyhole surgery for intracranial meningioma in a large case seriesPLOS ONE

Dear Dr. Kelly,

Thank you for submitting your manuscript to PLOS ONE. After careful consideration, we feel that it has merit but does not fully meet PLOS ONE’s publication criteria as it currently stands. Therefore, we invite you to submit a revised version of the manuscript that addresses the points raised during the review process.

We look forward to receiving your revised manuscript.

Kind regards,

Panagiotis Kerezoudis, M.D., M.S.

Academic Editor

PLOS ONE

Journal Requirements:

"I have read the journal's policy and the authors of this manuscript have the following competing interests: 

Dr. Kelly receives royalties from Mizuho Inc.

Dr. Barkhoudarian is a consultant for Vascular Technologies Inc. and Cerevasc Inc.

Dr. Sivakumar is a consultant for Stryker Corporation."

We note that you received funding from a commercial source: Mizuho Inc., Vascular Technologies Inc. , Cerevasc Inc. and Stryker Corporation

Reviewers' comments:

Reviewer's Responses to Questions

**Comments to the Author**

1. Is the manuscript technically sound, and do the data support the conclusions?

Reviewer #1: Partly

Reviewer #2: Partly

Reviewer #3: Yes

2. Has the statistical analysis been performed appropriately and rigorously? 

Reviewer #1: Yes

Reviewer #2: No

Reviewer #3: Yes

3. Have the authors made all data underlying the findings in their manuscript fully available?

Reviewer #1: No

Reviewer #2: No

Reviewer #3: Yes

4. Is the manuscript presented in an intelligible fashion and written in standard English?

Reviewer #1: Yes

Reviewer #2: Yes

Reviewer #3: Yes

5. Review Comments to the Author

Reviewer #1: The authors put forth a commendable effort to document their long-standing experience with minimally invasive approaches for intracranial meningiomas, which is a general trend in not only neurosurgery but other surgical subspecialties. The authors present a tremendous amount of primary data for the reader to review, which is important but becomes somewhat overwhelming for the reader. This is compounded somewhat by the weak connection between the two aims presented in the introduction and the methods/data with which they plan to assess these aims. Specific comments are below.

- endoscopic endonasal is fundamentally different really keyhole approach?

- comparable resection rates for endoscopic vs transcranial approaches? what's the comparison group?

Introduction

- There was overall good explanation of history leading up to current question/objective, but it's never really define what "minimally invasive" or "keyhole" surgery is, or whether that definition has changed over time. Some verbiage from the Discussion may be helpful here.

- Consider rewording line 95 to : Herein we report detailed outcomes "of our experience with minimally invasive approaches to intracranial meningiomas" including extent of resection, complications, LOS, performance status, readmissions, and resection and recurrence rates.

- LOS spell out length of stay pg 4 line 96

- The authors list two aims for the study, but it is never clearly stated what the criteria are in planning to answer those questions. Regarding the first question, ‘Is the benefit of a keyhole approach only on soft tissue & bone or also on the brain, and overall surgical recovery?', I assume many of the primary and some of the secondary outcomes (eg FLAIR changes) are aimed to answer this? How do you define overall surgical recovery (LOS, KPS)? What about soft tissue/bone? Regarding the second aim ‘Are keyhole approaches reasonable alternatives to traditional approaches? There is no mention of how this will be answered by the study; ie there is no comparison cohort, either from the institution's historical dataset or from the literature. The authors' data are placed into some context in the discussion, as compared to published literature, but this is not done in a systematic fashion as would be suggested by the second aim. If it, in fact, was not the intention of the paper, then consider rewording or removing this. Connecting the definitions of the outcomes with the aims should be included in the Methods.

Methods

- change tense to past in 2nd paragraph

- what other traditional skull base approaches do you include (line 118)? If no more, just remove this phrase.

Table 1-

under Falx, approach selection: do you mean trans-falcine instead of trans-tentorial?

Results:

What did you classify as an MIS operation in the same patients? Did you include patients that were multiply operated on for recurrences as two separate events? What about staged procedures? This should be made more clear in the Methods.

Discussion:

Was there any consideration of the COVID-19 pandemic as a contributor to shorter LOS in 2020-2021?

There is some debate as to whether endoscopic endonasal procedures are "minimally invasive" and categorized along with open craniotomies. Please comment.

The context for the current dataset was well set in this discussion, as it was compared to published series using traditional techniques; a range of percentages from previously published series in (page 19 second paragraph) would be helpful.

Reviewer #2: The authors report a single-center cohort study of 329 patients who underwent neurosurgical resection of an intracranial meningioma during the study period, 2008-2021, focusing on individuals treated with keyhole approaches (rather than conventional craniotomies), which were applied for the resection of 213 operations on 193 patients. Parameters studied include a range of operative details, outcomes, and complications, all reported through a predominantly descriptive approach to analysis.

Overall, the review is an well-presented and timely series that describes a relatively large experience and will likely be of interest to many in the neurosurgical readership. Several issues warrant further attention prior to consideration for publication:

1. The major issue with this paper, which is manifest in several distinct ways, is the overenthusiastic scope. In my view, the authors are trying to accomplish far too much with a single study, which has rendered the manuscript difficult to ready, digest, or consider for generalization and potential applications to one's own practice. I would strongly recommend consideration for a balkanization of the work into several daughter projects—perhaps one descriptive analysis of the group's minimally invasive cranial practice, followed by others focusing on smaller study questions, such as the role of "dense adherence" between the tumor and surrounding neurovascular structures as an predictor of a subtotal resection, or difference between conventional and minimally invasive outcomes

2. To that same end, candidly, the great majority of neurosurgeons would parse the endoscopic endonasal separately from the other keyhole approaches described—in particular, those with skull base expertise. Although both may be classified under the heading of "minimally invasive approaches," most of the factors that go into patient selection, operative planning, outcomes, etc. for endoscopic endonasal surgery are quite distinct from trans-cranial operations. The data from these patients would likely constitute an interesting stand-alone analysis; however, at the authors' discretion, leaving them in place as part of a larger descriptive-only series of "minimally invasive approaches" would not be unreasonable, (though any formal or informal comparisons would have to be stricken from such an analysis)

3. The authors have selected a very ambiguous definition for keyhole approaches, which would likely be subject to much debate within the skull base community. Additionally, they combine approaches with very different indications, risk profiles, etc. into a single analysis, for example the mini-pterional and retro-mastoid. Finally, many of the approaches sited as "keyhole" approaches are not practiced as such by all neurosurgeons, for example the trans-falcine approach, which refers to intra-cranial aspects of the dissection and approach rather than the craniotomy proper, and which can therefore be performed via a range of access ports of both large and small size. Considerable attention is required to better clarify and qualify these distinctions within the broader and highly diverse context of complex cranial surgery

4. Given the very heterogeneous sample, the results of any statistical testing conducted in this analysis would be subject to a very guarded interpretation at best. The authors should strongly consider removing all non-descriptive statistics from the main manuscript, and save them for separate subgroup analyses that would be reported separately, and where more homogeneity within the sample would potentially provide more meaningful interpretation

5. The decision to analyze GTR and NTR (defined as 90-99% resection) as a single subgroup is controversial, and not consistent with most contemporary data regarding the phenotypic behavior of incompletely resected meningiomas. A more robust approach would be to combine NTR and STR, given that the presence of any known residual solid tumor is well-known to increase the risk of disease recurrence, and the need for additional treatments in the future

Reviewer #3: This is a nice institutional report on use of minimally invasive approaches for intracranial meningiomas in 193 patients. They have used various approaches including endoscopic, minipterional, suboccipital approaches with or without the use of endoscopes and have shown great outcome in terms of resection rate, postoperative complications and clinical outcome at follow up. The choice and rationality of various approaches have been described in detail along with illustrative cases and operative videos. I would commend the authors for their sincere efforts in compiling such detailed report.

Few comments i would request the authors to address,

1. I feel the benefits of minimally approaches are partially offset by the high rate of reoperations due to residual disease. Overall reoperation rate of 22.1% including 32.4% in endoscopic endonasal cases, 21.9% in supraorbital cases seems high. Please comment.

2. Please explain the high rate of new/worsened FLAIR changes with suboccipital and retrosigmoid approaches.

3. The secondary analysis is poor. I do not see any relevant result or discussion on factors affecting the GTR or complications. Although there is a supplementary table (Table-2), not much has been discussed on it.

6. PLOS authors have the option to publish the peer review history of their article (what does this mean?). If published, this will include your full peer review and any attached files.

Reviewer #1: No

Reviewer #2: No

Reviewer #3: No

---

## [Author Response · Author response to Decision Letter 0]

13 Apr 2022

PONE-D-22-03249 Reviewer response: Critical appraisal of minimally invasive keyhole surgery for intracranial meningioma in a large case series at a single institution. Thakur et al

Please note: all page and line numbers refer to the “track changes” version of the manuscript.

Reviewer #1: The authors put forth a commendable effort to document their long-standing experience with minimally invasive approaches for intracranial meningiomas, which is a general trend in not only neurosurgery but other surgical subspecialties. The authors present a tremendous amount of primary data for the reader to review, which is important but becomes somewhat overwhelming for the reader. This is compounded somewhat by the weak connection between the two aims presented in the introduction and the methods/data with which they plan to assess these aims. Specific comments are below.

- endoscopic endonasal is fundamentally different really keyhole approach?

Response: 

We appreciate this comment but would like to stress that this paper is meant to describe a global approach to meningiomas using a 360-degree minimally invasive paradigm. As such, even though the endoscopic endonasal approach is not transcranial, it is fully aligned with the “keyhole concept” which stresses a “minimally invasive concept of microsurgery based on tailored, targeted, and direct microneurosurgical techniques” and is “a combination of modern microsurgical techniques, preoperative imaging, neuroendoscopy, and the modern concept of minimally invasive surgery”, as described by Lan et al (ref #21) in their consensus paper on keyhole neurosurgery. The fact that the endonasal approach is performed with an endoscope, and was used in 35% of all keyhole meningioma operations in this series, strongly argues that the endonasal route be included in any such analysis of minimally invasive keyhole surgery. This idea is exemplified by using a minimally invasive endonasal approach for managing specific skull base meningiomas as opposed to performing a traditional skull base approach such as an orbito-zygomatic approach with clinoidectomy for optic nerve decompression for a tuberculum sella meningioma. This point and the overall concept of the paper has now been further highlighted and detailed in the 2nd paragraph on page 4, lines 89-99 with additional supportive references added. We also note that we have already published peer-reviewed papers assessing minimally invasive keyhole meningioma surgery in elderly patients that included the endoscopic endonasal approach (Thakur et al 2020, ref #28) and in comparing two keyhole approaches, the endonasal versus supraorbital, for removal of tuberculum sella meningiomas (Mallari et al 2021, ref#27). 

- comparable resection rates for endoscopic vs transcranial approaches? what's the comparison group?

Response:

Excellent point raised. The comparison group is to prior publications for skull base meningiomas as we now highlight in the introduction (page 5 lines 105-116 and discussion. We add a new heading in the discussion on page 20: Complication rates, functional recovery and LOS compared to prior reports: to highlight our outcomes compared to prior publications. We also do the same on resection and remission rates in the discussion section on page 21 “Balancing goals of maximal tumor removal and complication avoidance:” This point of comparison to prior reports has also been clarified in the conclusion section of the abstract and in the introduction to reflect our intention. 

Introduction

- There was overall good explanation of history leading up to current question/objective, but it's never really defined what "minimally invasive" or "keyhole" surgery is, or whether that definition has changed over time. Some verbiage from the Discussion may be helpful here. 

Response:

We agree. We have added more detail about the “keyhole” surgical concept in the introduction (as noted above) in lines 89-99 on page 4. 

- Consider rewording line 95 to : Herein we report detailed outcomes "of our experience with minimally invasive approaches to intracranial meningiomas" including extent of resection, complications, LOS, performance status, readmissions, and resection and recurrence rates. - LOS spell out length of stay pg 4 line 96

Response:

Thank you. This change has been made (now on lines 105-107).

- The authors list two aims for the study, but it is never clearly stated what the criteria are in planning to answer those questions. Regarding the first question, ‘Is the benefit of a keyhole approach only on soft tissue & bone or also on the brain, and overall surgical recovery?', I assume many of the primary and some of the secondary outcomes (eg FLAIR changes) are aimed to answer this? How do you define overall surgical recovery (LOS, KPS)? What about soft tissue/bone? Regarding the second aim ‘Are keyhole approaches reasonable alternatives to traditional approaches? There is no mention of how this will be answered by the study; ie there is no comparison cohort, either from the institution's historical dataset or from the literature. The authors' data are placed into some context in the discussion, as compared to published literature, but this is not done in a systematic fashion as would be suggested by the second aim. If it, in fact, was not the intention of the paper, then consider rewording or removing this. Connecting the definitions of the outcomes with the aims should be included in the Methods.

Response: 

We agree that there was a “weak connection” between the two aims (we call them questions) posed in the introduction and have now attempted to clearly address this important issue. In this revision, we have now combined these two questions into one overarching Question on page 5 at end of introduction (lines 114-116): ‘What is the benefit of keyhole surgery on neurological and overall surgical recovery, and can such minimally invasive approaches be considered reasonable alternatives to traditional approaches?’ We have also more clearly indicated how this question is answered in the introduction, methods, results and discussion. In the introduction (lines 105-112), we now clarify that this question is answered by our clinical outcome measures (both primary and secondary as defined in the methods). This point has been highlighted in the methodology under Outcome and Statistical Analysis (page 10, lines 187-189). We then address this question in the discussion further in the first paragraph (page 18, lines 306-309) and in section heading (page 20): Complication rates, functional recovery and LOS compared to prior reports, by comparing our overall results (especially focused on complications and LOS) with prior publications using traditional skull base approaches, and similarly on extent of resection and recurrence, in the next discussion section (page 21): Balancing goals of maximal tumor removal and complication avoidance. 

Methods

- change tense to past in 2nd paragraph

- what other traditional skull base approaches do you include (line 118)? If no more, just remove this phrase.

Response:

This tense has been changed to past tense and the traditional skull base approaches section has been revised as requested in the same paragraph. 

Table 1-

under Falx, approach selection: do you mean trans-falcine instead of trans-tentorial?

Response:

Thank you for noting this; we did mean trans-falcine. This has been corrected.

Results:

What did you classify as an MIS operation in the same patients? Did you include patients that were multiply operated on for recurrences as two separate events? What about staged procedures? This should be made more clear in the Methods. 

Response:

This is an important point. All patients included in the keyhole cohort had an MIS approach. Being reoperated on for a recurrence is thus considered an outcome measure which we have in the results section as a secondary outcome (page 17, lines 291-296). Our overall recurrence rate was 13.9% (27/193) and of these 6 (3%) patients had repeat surgery with or without SRT or SRS. Regarding staged procedures, of the 4 patients, only one had a traditional pterional craniotomy combined with an endonasal approach (as detailed now page 11 lines 221-222). 

Discussion:

Was there any consideration of the COVID-19 pandemic as a contributor to shorter LOS in 2020-2021?

There is some debate as to whether endoscopic endonasal procedures are "minimally invasive" and categorized along with open craniotomies. Please comment. 

Response:

The reviewer raises an important point regarding the impact of the pandemic on LOS. In fact, we have published on this very topic and ICU utilization for all brain tumor patients (Mallari et al ref # 35 published in PLOS One in 2021). Even though COVID did play a role in considering earlier discharge, our use of MIS keyhole approaches began more than a decade before the pandemic allowing us to have a shorter LOS compared to other similar meningioma skull base cohort series. However, we now point out in the discussion (page 21, lines 366-376) that the median LOS in the last 71 cases was also likely further encouraged by our efforts to further reduce LOS and ICU given the resource and bed demands of the pandemic; we briefly describe the results of this paper. 

Regarding whether endoscopic endonasal should be considered a “minimally invasive” approach and categorized with transcranial approaches, we have expanded upon this point in the introduction (page 4) and the discussion (page 18) as noted above. We emphasize that by including the endonasal route, we are endorsing a 360-degree minimally invasive keyhole approach that argues strongly to use the optimal route that minimizes brain and cranial nerve manipulation, promotes a rapid recovery yet allows maximal safe tumor removal. Thus, we respectfully argue that the endonasal approach be included in this “minimally invasive” armamentarium as we have already done in multiple prior reports (Thakur et al 2020, ref #28; Mallari et al 2021, ref #27; Mallari et al, 2021, ref #35). 

The context for the current dataset was well set in this discussion, as it was compared to published series using traditional techniques; a range of percentages from previously published series in (page 19 second paragraph) would be helpful. 

Response:

Thanks for this suggestion. On page 20, we had already provided a range of percentages on neurological and other complications, LOS and readmission rates. We also provide resection rates from the literature for different meningioma locations on pages 21-22 (lines 386-392). 

Reviewer #2: The authors report a single-center cohort study of 329 patients who underwent neurosurgical resection of an intracranial meningioma during the study period, 2008-2021, focusing on individuals treated with keyhole approaches (rather than conventional craniotomies), which were applied for the resection of 213 operations on 193 patients. Parameters studied include a range of operative details, outcomes, and complications, all reported through a predominantly descriptive approach to analysis.

Overall, the review is an well-presented and timely series that describes a relatively large experience and will likely be of interest to many in the neurosurgical readership. Several issues warrant further attention prior to consideration for publication:

1. The major issue with this paper, which is manifest in several distinct ways, is the overenthusiastic scope. In my view, the authors are trying to accomplish far too much with a single study, which has rendered the manuscript difficult to ready, digest, or consider for generalization and potential applications to one's own practice. I would strongly recommend consideration for a balkanization of the work into several daughter projects—perhaps one descriptive analysis of the group's minimally invasive cranial practice, followed by others focusing on smaller study questions, such as the role of "dense adherence" between the tumor and surrounding neurovascular structures as an predictor of a subtotal resection, or difference between conventional and minimally invasive outcomes. 

Response:

We appreciate the reviewer’s critique and agree that we have been expansive and inclusive in terms of this large meningioma data set. However, we would argue that in many if not most prior publications on meningioma surgery, a comprehensive review is not undertaken. For example, many papers will focus only on extent of resection and recurrence, while others will focus on complications, length of stay and/or quality of life. We suggest that to answer our central question (Introduction page 5), ‘What is the benefit of keyhole surgery on neurological and overall surgical recovery, and can such minimally invasive approaches be considered reasonable alternatives to traditional approaches?’ the entire patient experience needs to be considered to understand how patients fare and thus help derive an optimal approach for each patient. So, in that context we have attempted to look at our overall approach for meningiomas including all surgical approaches (including endoscopic endonasal) and look at our detailed outcomes. Only by completing such a comprehensive analysis, can the total impact of this approach be assessed. Thus, as we emphasize in the discussion, having a high rate of GTR or NTR but with an equally high rate of permanent neurological complications (as has been shown in many articles detailing outcomes of traditional skull base approaches) is not what the neurosurgical community should be striving for. While we note that we have some of the lowest complication rates and LOS in the literature, we acknowledge that ours is only one experience, and we need longer follow-up of this keyhole paradigm.

Additionally, we respectfully request that we not “balkanize” our results into smaller papers, and in fact, we have already done four such “balkanized” papers including: endonasal versus supraorbital for tuberculum sella meningiomas (Mallari et al, 2021, ref #27); invasive parasellar meningiomas treated by endoscopic endonasal route (Sivakumar et al 2019, ref #32 ), supraorbital versus minipterional approach for all tumors (Avery et al 2021; ref #34), and endoscope-assisted transfalcine approach (Barkhoudarian et al, 2017, ref #29). Thus, as we indicated on page 4 of introduction, this paper and analysis is trying to take a broader perspective on overall meningioma management focusing on the potential benefits of this 360-degree keyhole paradigm. 

2. To that same end, candidly, the great majority of neurosurgeons would parse the endoscopic endonasal separately from the other keyhole approaches described—in particular, those with skull base expertise. Although both may be classified under the heading of "minimally invasive approaches," most of the factors that go into patient selection, operative planning, outcomes, etc. for endoscopic endonasal surgery are quite distinct from trans-cranial operations. The data from these patients would likely constitute an interesting stand-alone analysis; however, at the authors' discretion, leaving them in place as part of a larger descriptive-only series of "minimally invasive approaches" would not be unreasonable, (though any formal or informal comparisons would have to be stricken from such an analysis) 

Response:

We appreciate this sentiment but as detailed above in prior responses, we have already written such stand-alone papers comparing keyhole approaches. For example, in managing tuberculum sella meningiomas by endonasal vs supraorbital route (Mallari et al, 2021, ref #27). In that paper we emphasize that the endonasal and supraorbital approaches are complimentary keyhole approaches and need to be considered together as to which is the optimal approach. Thus, we respectfully disagree and instead believe that including endoscopic endonasal in the keyhole paradigm and armamentarium is appropriate and is needed to help further advance the field of skull base surgery. 

3. The authors have selected a very ambiguous definition for keyhole approaches, which would likely be subject to much debate within the skull base community. Additionally, they combine approaches with very different indications, risk profiles, etc. into a single analysis, for example the mini-pterional and retro-mastoid. Finally, many of the approaches sited as "keyhole" approaches are not practiced as such by all neurosurgeons, for example the trans-falcine approach, which refers to intra-cranial aspects of the dissection and approach rather than the craniotomy proper, and which can therefore be performed via a range of access ports of both large and small size. Considerable attention is required to better clarify and qualify these distinctions within the broader and highly diverse context of complex cranial surgery 

Response:

We appreciate the concern about possible ambiguity of the definition of the keyhole approach. However, as we have attempted to emphasize in other responses to Reviewer #1 and #2, and in the revised manuscript methods section, we have clarified our definition of “keyhole approach”. We believe that all 6 approaches we include meet the criteria as defined in the consensus statement for keyhole concept (Lan et al 2019, ref#21) and in our prior papers. Inclusion of the trans-falcine approach for parafalcine meningiomas does introduce heterogeneity in the data albeit not significantly since the patient population in this group is only 2% of the cohort. Additionally, the parafalcine meningiomas included in the analysis are underlying eloquent cortex where use of a port would not be appropriate as it would undoubtedly disrupt the eloquent cortex one is trying save. Importantly, the transfalcine approach for removing falx meningiomas cannot be safely done ( in our opinion) without endoscopic angled visualization which allows one to achieve complete meningioma removal without brain retraction. Applying endoscopy (when needed) is one of the key technological tenets of MIS keyhole brain tumor surgery as stressed by Lan et al (2019). 

4. Given the very heterogeneous sample, the results of any statistical testing conducted in this analysis would be subject to a very guarded interpretation at best. The authors should strongly consider removing all non-descriptive statistics from the main manuscript, and save them for separate subgroup analyses that would be reported separately, and where more homogeneity within the sample would potentially provide more meaningful interpretation 

Response:

Thank you for the suggestion. The heterogeneity is certainly a limitation of the study which has been added in the limitation section (page 23, line 413). However, we have used Binomial Multivariate Regression analysis to minimize confounding factors, and which as requested by the reviewers is reported only in the supplemental data and not in the main manuscript. 

5. The decision to analyze GTR and NTR (defined as 90-99% resection) as a single subgroup is controversial, and not consistent with most contemporary data regarding the phenotypic behavior of incompletely resected meningiomas. A more robust approach would be to combine NTR and STR, given that the presence of any known residual solid tumor is well-known to increase the risk of disease recurrence, and the need for additional treatments in the future 

Response:

We agree with the reviewer that our grouping approach of GTR with NTR is somewhat controversial, however we emphasize throughout the paper that our surgical goal is always “maximal safe tumor removal” and we do show the GTR rates for the different approaches and tumor locations in both Tables 2 and 3. Given the highly invasive skull base tumor population, the frequent finding of vascular encasement, and that many patients had prior surgery and/or radiation, settling for NTR in many cases is a reasonable goal. Thus, we respectfully request to leave the GTR/NTR grouping as is, believing that keeping this grouping aligns with an overall minimally invasive paradigm that also strives for a low complication rate, short LOS and high QOL. Furthermore, although the results are limited by relatively shorter follow up, our tumor control rates are comparable to previously published rates as noted in the discussion (page 21, lines 381-385).

Reviewer #3: This is a nice institutional report on use of minimally invasive approaches for intracranial meningiomas in 193 patients. They have used various approaches including endoscopic, minipterional, suboccipital approaches with or without the use of endoscopes and have shown great outcome in terms of resection rate, postoperative complications and clinical outcome at follow up. The choice and rationality of various approaches have been described in detail along with illustrative cases and operative videos. I would commend the authors for their sincere efforts in compiling such detailed report.

Response:

We appreciate the acknowledgement. Thank you. 

Few comments I would request the authors to address,

1. I feel the benefits of minimally approaches are partially offset by the high rate of reoperations due to residual disease. Overall reoperation rate of 22.1% including 32.4% in endoscopic endonasal cases, 21.9% in supraorbital cases seems high. Please comment. 

Response:

Thank you for this comment and sorry for the confusion. The actual reoperation rate for tumor progression was 3.1%. To clarify, those rates of 22.1% (overall), 32.4% (endonasal) and 21.9 (supraorbital) listed in Table 2 indicate patients who came to us after having prior surgery. We have clarified this point in Table 2. The overall rate of tumor progression for the entire cohort during the study period (as we state in the results section on page 17, lines 291-293): Long-term tumor progression or recurrence) was 13.9% (27/193 patients) including 3.1% (6 patients) who had repeat surgery with or without SRS or SRT. 

2. Please explain the high rate of new/worsened FLAIR changes with suboccipital and retrosigmoid approaches. 

Response:

The higher rate of FLAIR changes at 3 months (12.5%) for the retromastoid approach may be related to the fact that this approach for many such meningiomas (CP angle and petroclival), the cerebellar hemisphere partially obstructs tumor access, and despite excellent neuroanesthesia, positioning and opening of cisterns for CSF creating a relaxed posterior fossa, some manipulation of the cerebellar hemisphere is unavoidable which may lead to new FLAIR changes in some cases. That said, since there is little data on early postoperative FLAIR changes in skull base meningioma series, we feel our rate of FLAIR changes is difficult to compare to prior reports. Notably, persistent FLAIR changes at 3 months were noted only in 5% of patients and were quite small and did not seem to correlate with lasting neurological deficits (Table 2). Overall, these rates appear to be quite low (including for retromastoid approach) compared for example to Bander et al (ref #46) which had a mean post-operative FLAIR volume of 8.3 cm3 in patients who had traditional transcranial removal of tuberculum meningiomas. 

3. The secondary analysis is poor. I do not see any relevant result or discussion on factors affecting the GTR or complications. Although there is a supplementary table (Table-2), not much has been discussed on it. 

Response:

Thank you for the suggestion. We highlight the factors influencing resectability in Table 3 and supplemental Table 2. The most important information in supplemental Table 2 is emphasizing that Invasion to CS-MC-Orbit-ITF/ Adherence to neurovascular structures strongly predicted non-GTR. This point is now addressed in a new section of the discussion (page 22, lines 404-411).

---

## [Decision Letter · Decision Letter 1]

24 May 2022

PONE-D-22-03249R1Critical appraisal of minimally invasive keyhole surgery for intracranial meningioma in a large case seriesPLOS ONE

Dear Dr. Kelly,

Thank you for submitting your manuscript to PLOS ONE. After careful consideration, we feel that it has merit but does not fully meet PLOS ONE’s publication criteria as it currently stands. Therefore, we invite you to submit a revised version of the manuscript that addresses the points raised during the review process.

Dear Dr. Kelly,  We appreciate your thoughtful response to the reviewers. I agree with reviewer#2 that some of the concerns have not been adequately addressed and no meaningful revisions have been made to the manuscript. Please ensure to address the original comments raised by the reviewer#2. 

We look forward to receiving your revised manuscript.

Kind regards,

Panagiotis Kerezoudis, M.D., M.S.

Academic Editor

PLOS ONE

Reviewers' comments:

Reviewer's Responses to Questions

**Comments to the Author**

1. If the authors have adequately addressed your comments raised in a previous round of review and you feel that this manuscript is now acceptable for publication, you may indicate that here to bypass the “Comments to the Author” section, enter your conflict of interest statement in the “Confidential to Editor” section, and submit your "Accept" recommendation.

Reviewer #1: All comments have been addressed

Reviewer #2: (No Response)

Reviewer #3: All comments have been addressed

2. Is the manuscript technically sound, and do the data support the conclusions?

Reviewer #1: Yes

Reviewer #2: Partly

Reviewer #3: Yes

3. Has the statistical analysis been performed appropriately and rigorously? 

Reviewer #1: Yes

Reviewer #2: No

Reviewer #3: Yes

4. Have the authors made all data underlying the findings in their manuscript fully available?

Reviewer #1: Yes

Reviewer #2: No

Reviewer #3: Yes

5. Is the manuscript presented in an intelligible fashion and written in standard English?

Reviewer #1: Yes

Reviewer #2: Yes

Reviewer #3: Yes

6. Review Comments to the Author

Reviewer #1: All comments have been appropriately addressed, particularly the methodology, the rationale for including endoscopic endonasal approaches, and their comparison groups. The manuscript is now fit for publication.

Reviewer #2: The authors have made no meaningful effort to address the concerns raised; rather, their responses are prosaic, tangential attempts to justify why they feel that these critiques do not merit further attention. Fortunately, as they have handily detailed via several proffered citations, the current work in fact represents a redundant synthesis of previously published data. Taken in that light, and given that it remains essentially unrevised as compared to the initial submission, it is difficult to justify recommending publication at this time.

Reviewer #3: The authors have addressed the queries and made revisions in the manuscript accordingly. I would recommend for publication

---

## [Author Response · Author response to Decision Letter 1]

30 Jun 2022

PONE-D-22-03249 Reviewer response: Critical appraisal of minimally invasive keyhole surgery for intracranial meningioma in a large case series at a single institution. Thakur et al

Please note: all page and line numbers refer to the “track changes” version of the manuscript.

Reviewer #1: The authors put forth a commendable effort to document their long-standing experience with minimally invasive approaches for intracranial meningiomas, which is a general trend in not only neurosurgery but other surgical subspecialties. The authors present a tremendous amount of primary data for the reader to review, which is important but becomes somewhat overwhelming for the reader. This is compounded somewhat by the weak connection between the two aims presented in the introduction and the methods/data with which they plan to assess these aims. Specific comments are below.

- endoscopic endonasal is fundamentally different really keyhole approach?

Response: 

We appreciate this comment but would like to stress that this paper is meant to describe a global approach to meningiomas using a 360-degree minimally invasive paradigm. As such, even though the endoscopic endonasal approach is not transcranial, it is fully aligned with the “keyhole concept” which stresses a “minimally invasive concept of microsurgery based on tailored, targeted, and direct microneurosurgical techniques” and is “a combination of modern microsurgical techniques, preoperative imaging, neuroendoscopy, and the modern concept of minimally invasive surgery”, as described by Lan et al (ref #21) in their consensus paper on keyhole neurosurgery. The fact that the endonasal approach is performed with an endoscope, and was used in 35% of all keyhole meningioma operations in this series, strongly argues that the endonasal route be included in any such analysis of minimally invasive keyhole surgery. This idea is exemplified by using a minimally invasive endonasal approach for managing specific skull base meningiomas as opposed to performing a traditional skull base approach such as an orbito-zygomatic approach with clinoidectomy for optic nerve decompression for a tuberculum sella meningioma. This point and the overall concept of the paper has now been further highlighted and detailed in the 2nd paragraph on page 4, lines 89-99 with additional supportive references added. We also note that we have already published peer-reviewed papers assessing minimally invasive keyhole meningioma surgery in elderly patients that included the endoscopic endonasal approach (Thakur et al 2020, ref #28) and in comparing two keyhole approaches, the endonasal versus supraorbital, for removal of tuberculum sella meningiomas (Mallari et al 2021, ref#27). 

- comparable resection rates for endoscopic vs transcranial approaches? what's the comparison group?

Response:

Excellent point raised. The comparison group is to prior publications for skull base meningiomas as we now highlight in the introduction (page 5 lines 105-116 and discussion. We add a new heading in the discussion on page 20: Complication rates, functional recovery and LOS compared to prior reports: to highlight our outcomes compared to prior publications. We also do the same on resection and remission rates in the discussion section on page 21 “Balancing goals of maximal tumor removal and complication avoidance:” This point of comparison to prior reports has also been clarified in the conclusion section of the abstract and in the introduction to reflect our intention. 

Introduction

- There was overall good explanation of history leading up to current question/objective, but it's never really defined what "minimally invasive" or "keyhole" surgery is, or whether that definition has changed over time. Some verbiage from the Discussion may be helpful here. 

Response:

We agree. We have added more detail about the “keyhole” surgical concept in the introduction (as noted above) in lines 89-99 on page 4. 

- Consider rewording line 95 to : Herein we report detailed outcomes "of our experience with minimally invasive approaches to intracranial meningiomas" including extent of resection, complications, LOS, performance status, readmissions, and resection and recurrence rates. - LOS spell out length of stay pg 4 line 96

Response:

Thank you. This change has been made (now on lines 105-107).

- The authors list two aims for the study, but it is never clearly stated what the criteria are in planning to answer those questions. Regarding the first question, ‘Is the benefit of a keyhole approach only on soft tissue & bone or also on the brain, and overall surgical recovery?', I assume many of the primary and some of the secondary outcomes (eg FLAIR changes) are aimed to answer this? How do you define overall surgical recovery (LOS, KPS)? What about soft tissue/bone? Regarding the second aim ‘Are keyhole approaches reasonable alternatives to traditional approaches? There is no mention of how this will be answered by the study; ie there is no comparison cohort, either from the institution's historical dataset or from the literature. The authors' data are placed into some context in the discussion, as compared to published literature, but this is not done in a systematic fashion as would be suggested by the second aim. If it, in fact, was not the intention of the paper, then consider rewording or removing this. Connecting the definitions of the outcomes with the aims should be included in the Methods.

Response: 

We agree that there was a “weak connection” between the two aims (we call them questions) posed in the introduction and have now attempted to clearly address this important issue. In this revision, we have now combined these two questions into one overarching Question on page 5 at end of introduction (lines 114-116): ‘What is the benefit of keyhole surgery on neurological and overall surgical recovery, and can such minimally invasive approaches be considered reasonable alternatives to traditional approaches?’ We have also more clearly indicated how this question is answered in the introduction, methods, results and discussion. In the introduction (lines 105-112), we now clarify that this question is answered by our clinical outcome measures (both primary and secondary as defined in the methods). This point has been highlighted in the methodology under Outcome and Statistical Analysis (page 10, lines 187-189). We then address this question in the discussion further in the first paragraph (page 18, lines 306-309) and in section heading (page 20): Complication rates, functional recovery and LOS compared to prior reports, by comparing our overall results (especially focused on complications and LOS) with prior publications using traditional skull base approaches, and similarly on extent of resection and recurrence, in the next discussion section (page 21): Balancing goals of maximal tumor removal and complication avoidance. 

Methods

- change tense to past in 2nd paragraph

- what other traditional skull base approaches do you include (line 118)? If no more, just remove this phrase.

Response:

This tense has been changed to past tense and the traditional skull base approaches section has been revised as requested in the same paragraph. 

Table 1-

under Falx, approach selection: do you mean trans-falcine instead of trans-tentorial?

Response:

Thank you for noting this; we did mean trans-falcine. This has been corrected.

Results:

What did you classify as an MIS operation in the same patients? Did you include patients that were multiply operated on for recurrences as two separate events? What about staged procedures? This should be made more clear in the Methods. 

Response:

This is an important point. All patients included in the keyhole cohort had an MIS approach. Being reoperated on for a recurrence is thus considered an outcome measure which we have in the results section as a secondary outcome (page 17, lines 291-296). Our overall recurrence rate was 13.9% (27/193) and of these 6 (3%) patients had repeat surgery with or without SRT or SRS. Regarding staged procedures, of the 4 patients, only one had a traditional pterional craniotomy combined with an endonasal approach (as detailed now page 11 lines 221-222). 

Discussion:

Was there any consideration of the COVID-19 pandemic as a contributor to shorter LOS in 2020-2021?

There is some debate as to whether endoscopic endonasal procedures are "minimally invasive" and categorized along with open craniotomies. Please comment. 

Response:

The reviewer raises an important point regarding the impact of the pandemic on LOS. In fact, we have published on this very topic and ICU utilization for all brain tumor patients (Mallari et al ref # 35 published in PLOS One in 2021). Even though COVID did play a role in considering earlier discharge, our use of MIS keyhole approaches began more than a decade before the pandemic allowing us to have a shorter LOS compared to other similar meningioma skull base cohort series. However, we now point out in the discussion (page 21, lines 366-376) that the median LOS in the last 71 cases was also likely further encouraged by our efforts to further reduce LOS and ICU given the resource and bed demands of the pandemic; we briefly describe the results of this paper. 

Regarding whether endoscopic endonasal should be considered a “minimally invasive” approach and categorized with transcranial approaches, we have expanded upon this point in the introduction (page 4) and the discussion (page 18) as noted above. We emphasize that by including the endonasal route, we are endorsing a 360-degree minimally invasive keyhole approach that argues strongly to use the optimal route that minimizes brain and cranial nerve manipulation, promotes a rapid recovery yet allows maximal safe tumor removal. Thus, we respectfully argue that the endonasal approach be included in this “minimally invasive” armamentarium as we have already done in multiple prior reports (Thakur et al 2020, ref #28; Mallari et al 2021, ref #27; Mallari et al, 2021, ref #35). 

The context for the current dataset was well set in this discussion, as it was compared to published series using traditional techniques; a range of percentages from previously published series in (page 19 second paragraph) would be helpful. 

Response:

Thanks for this suggestion. On page 20, we had already provided a range of percentages on neurological and other complications, LOS and readmission rates. We also provide resection rates from the literature for different meningioma locations on pages 21-22 (lines 386-392). 

Reviewer #2: The authors report a single-center cohort study of 329 patients who underwent neurosurgical resection of an intracranial meningioma during the study period, 2008-2021, focusing on individuals treated with keyhole approaches (rather than conventional craniotomies), which were applied for the resection of 213 operations on 193 patients. Parameters studied include a range of operative details, outcomes, and complications, all reported through a predominantly descriptive approach to analysis.

Overall, the review is an well-presented and timely series that describes a relatively large experience and will likely be of interest to many in the neurosurgical readership. Several issues warrant further attention prior to consideration for publication:

1. The major issue with this paper, which is manifest in several distinct ways, is the overenthusiastic scope. In my view, the authors are trying to accomplish far too much with a single study, which has rendered the manuscript difficult to ready, digest, or consider for generalization and potential applications to one's own practice. I would strongly recommend consideration for a balkanization of the work into several daughter projects—perhaps one descriptive analysis of the group's minimally invasive cranial practice, followed by others focusing on smaller study questions, such as the role of "dense adherence" between the tumor and surrounding neurovascular structures as an predictor of a subtotal resection, or difference between conventional and minimally invasive outcomes. 

Response:

We appreciate the reviewer’s critique and agree that we have been expansive and inclusive in terms of this large meningioma data set. However, we would argue that in many if not most prior publications on meningioma surgery, a comprehensive review is not undertaken. For example, many papers will focus only on extent of resection and recurrence, while others will focus on complications, length of stay and/or quality of life. We suggest that to answer our central question (Introduction page 5), ‘What is the benefit of keyhole surgery on neurological and overall surgical recovery, and can such minimally invasive approaches be considered reasonable alternatives to traditional approaches?’ the entire patient experience needs to be considered to understand how patients fare and thus help derive an optimal approach for each patient. So, in that context we have attempted to look at our overall approach for meningiomas including all surgical approaches (including endoscopic endonasal) and look at our detailed outcomes. Only by completing such a comprehensive analysis, can the total impact of this approach be assessed. Thus, as we emphasize in the discussion, having a high rate of GTR or NTR but with an equally high rate of permanent neurological complications (as has been shown in many articles detailing outcomes of traditional skull base approaches) is not what the neurosurgical community should be striving for. While we note that we have some of the lowest complication rates and LOS in the literature, we acknowledge that ours is only one experience, and we need longer follow-up of this keyhole paradigm.

Additionally, we respectfully request that we not “balkanize” our results into smaller papers, and in fact, we have already done four such “balkanized” papers including: endonasal versus supraorbital for tuberculum sella meningiomas (Mallari et al, 2021, ref #27); invasive parasellar meningiomas treated by endoscopic endonasal route (Sivakumar et al 2019, ref #32 ), supraorbital versus minipterional approach for all tumors (Avery et al 2021; ref #34), and endoscope-assisted transfalcine approach (Barkhoudarian et al, 2017, ref #29). Thus, as we indicated on page 4 of introduction, this paper and analysis is trying to take a broader perspective on overall meningioma management focusing on the potential benefits of this 360-degree keyhole paradigm. 

2. To that same end, candidly, the great majority of neurosurgeons would parse the endoscopic endonasal separately from the other keyhole approaches described—in particular, those with skull base expertise. Although both may be classified under the heading of "minimally invasive approaches," most of the factors that go into patient selection, operative planning, outcomes, etc. for endoscopic endonasal surgery are quite distinct from trans-cranial operations. The data from these patients would likely constitute an interesting stand-alone analysis; however, at the authors' discretion, leaving them in place as part of a larger descriptive-only series of "minimally invasive approaches" would not be unreasonable, (though any formal or informal comparisons would have to be stricken from such an analysis) 

Response:

We appreciate this sentiment but as detailed above in prior responses, we have already written such stand-alone papers comparing keyhole approaches. For example, in managing tuberculum sella meningiomas by endonasal vs supraorbital route (Mallari et al, 2021, ref #27). In that paper we emphasize that the endonasal and supraorbital approaches are complimentary keyhole approaches and need to be considered together as to which is the optimal approach. Thus, we respectfully disagree and instead believe that including endoscopic endonasal in the keyhole paradigm and armamentarium is appropriate and is needed to help further advance the field of skull base surgery. 

3. The authors have selected a very ambiguous definition for keyhole approaches, which would likely be subject to much debate within the skull base community. Additionally, they combine approaches with very different indications, risk profiles, etc. into a single analysis, for example the mini-pterional and retro-mastoid. Finally, many of the approaches sited as "keyhole" approaches are not practiced as such by all neurosurgeons, for example the trans-falcine approach, which refers to intra-cranial aspects of the dissection and approach rather than the craniotomy proper, and which can therefore be performed via a range of access ports of both large and small size. Considerable attention is required to better clarify and qualify these distinctions within the broader and highly diverse context of complex cranial surgery 

Response:

We appreciate the concern about possible ambiguity of the definition of the keyhole approach. However, as we have attempted to emphasize in other responses to Reviewer #1 and #2, and in the revised manuscript methods section, we have clarified our definition of “keyhole approach”. We believe that all 6 approaches we include meet the criteria as defined in the consensus statement for keyhole concept (Lan et al 2019, ref#21) and in our prior papers. Inclusion of the trans-falcine approach for parafalcine meningiomas does introduce heterogeneity in the data albeit not significantly since the patient population in this group is only 2% of the cohort. Additionally, the parafalcine meningiomas included in the analysis are underlying eloquent cortex where use of a port would not be appropriate as it would undoubtedly disrupt the eloquent cortex one is trying save. Importantly, the transfalcine approach for removing falx meningiomas cannot be safely done ( in our opinion) without endoscopic angled visualization which allows one to achieve complete meningioma removal without brain retraction. Applying endoscopy (when needed) is one of the key technological tenets of MIS keyhole brain tumor surgery as stressed by Lan et al (2019). 

4. Given the very heterogeneous sample, the results of any statistical testing conducted in this analysis would be subject to a very guarded interpretation at best. The authors should strongly consider removing all non-descriptive statistics from the main manuscript, and save them for separate subgroup analyses that would be reported separately, and where more homogeneity within the sample would potentially provide more meaningful interpretation 

Response:

Thank you for the suggestion. The heterogeneity is certainly a limitation of the study which has been added in the limitation section (page 23, line 413). However, we have used Binomial Multivariate Regression analysis to minimize confounding factors, and which as requested by the reviewers is reported only in the supplemental data and not in the main manuscript. 

5. The decision to analyze GTR and NTR (defined as 90-99% resection) as a single subgroup is controversial, and not consistent with most contemporary data regarding the phenotypic behavior of incompletely resected meningiomas. A more robust approach would be to combine NTR and STR, given that the presence of any known residual solid tumor is well-known to increase the risk of disease recurrence, and the need for additional treatments in the future 

Response:

We agree with the reviewer that our grouping approach of GTR with NTR is somewhat controversial, however we emphasize throughout the paper that our surgical goal is always “maximal safe tumor removal” and we do show the GTR rates for the different approaches and tumor locations in both Tables 2 and 3. Given the highly invasive skull base tumor population, the frequent finding of vascular encasement, and that many patients had prior surgery and/or radiation, settling for NTR in many cases is a reasonable goal. Thus, we respectfully request to leave the GTR/NTR grouping as is, believing that keeping this grouping aligns with an overall minimally invasive paradigm that also strives for a low complication rate, short LOS and high QOL. Furthermore, although the results are limited by relatively shorter follow up, our tumor control rates are comparable to previously published rates as noted in the discussion (page 21, lines 381-385).

Reviewer #3: This is a nice institutional report on use of minimally invasive approaches for intracranial meningiomas in 193 patients. They have used various approaches including endoscopic, minipterional, suboccipital approaches with or without the use of endoscopes and have shown great outcome in terms of resection rate, postoperative complications and clinical outcome at follow up. The choice and rationality of various approaches have been described in detail along with illustrative cases and operative videos. I would commend the authors for their sincere efforts in compiling such detailed report.

Response:

We appreciate the acknowledgement. Thank you. 

Few comments I would request the authors to address,

1. I feel the benefits of minimally approaches are partially offset by the high rate of reoperations due to residual disease. Overall reoperation rate of 22.1% including 32.4% in endoscopic endonasal cases, 21.9% in supraorbital cases seems high. Please comment. 

Response:

Thank you for this comment and sorry for the confusion. The actual reoperation rate for tumor progression was 3.1%. To clarify, those rates of 22.1% (overall), 32.4% (endonasal) and 21.9 (supraorbital) listed in Table 2 indicate patients who came to us after having prior surgery. We have clarified this point in Table 2. The overall rate of tumor progression for the entire cohort during the study period (as we state in the results section on page 17, lines 291-293): Long-term tumor progression or recurrence) was 13.9% (27/193 patients) including 3.1% (6 patients) who had repeat surgery with or without SRS or SRT. 

2. Please explain the high rate of new/worsened FLAIR changes with suboccipital and retrosigmoid approaches. 

Response:

The higher rate of FLAIR changes at 3 months (12.5%) for the retromastoid approach may be related to the fact that this approach for many such meningiomas (CP angle and petroclival), the cerebellar hemisphere partially obstructs tumor access, and despite excellent neuroanesthesia, positioning and opening of cisterns for CSF creating a relaxed posterior fossa, some manipulation of the cerebellar hemisphere is unavoidable which may lead to new FLAIR changes in some cases. That said, since there is little data on early postoperative FLAIR changes in skull base meningioma series, we feel our rate of FLAIR changes is difficult to compare to prior reports. Notably, persistent FLAIR changes at 3 months were noted only in 5% of patients and were quite small and did not seem to correlate with lasting neurological deficits (Table 2). Overall, these rates appear to be quite low (including for retromastoid approach) compared for example to Bander et al (ref #46) which had a mean post-operative FLAIR volume of 8.3 cm3 in patients who had traditional transcranial removal of tuberculum meningiomas. 

3. The secondary analysis is poor. I do not see any relevant result or discussion on factors affecting the GTR or complications. Although there is a supplementary table (Table-2), not much has been discussed on it. 

Response:

Thank you for the suggestion. We highlight the factors influencing resectability in Table 3 and supplemental Table 2. The most important information in supplemental Table 2 is emphasizing that Invasion to CS-MC-Orbit-ITF/ Adherence to neurovascular structures strongly predicted non-GTR. This point is now addressed in a new section of the discussion (page 22, lines 404-411).

---

## [Editor Report · Decision Letter 2]

4 Jul 2022

Critical appraisal of minimally invasive keyhole surgery for intracranial meningioma in a large case series

PONE-D-22-03249R2

Dear Dr. Kelly,

We’re pleased to inform you that your manuscript has been judged scientifically suitable for publication and will be formally accepted for publication once it meets all outstanding technical requirements.

Kind regards,

Panagiotis Kerezoudis, M.D., M.S.

Academic Editor

PLOS ONE
---

## [Editor Report · Acceptance letter]

18 Jul 2022

PONE-D-22-03249R2 

Critical appraisal of minimally invasive keyhole surgery for intracranial meningioma in a large case series 

Dear Dr. Kelly:

I'm pleased to inform you that your manuscript has been deemed suitable for publication in PLOS ONE. Congratulations! Your manuscript is now with our production department. 

Kind regards, 

on behalf of

Dr. Panagiotis Kerezoudis 

Academic Editor

PLOS ONE